# Evaluating Known Zika Virus NS2B-NS3 Protease Inhibitor Scaffolds via In Silico Screening and Biochemical Assays

**DOI:** 10.3390/ph16091319

**Published:** 2023-09-19

**Authors:** Lucianna H. Santos, Rafael E. O. Rocha, Diego L. Dias, Beatriz M. R. M. Ribeiro, Mateus Sá M. Serafim, Jônatas S. Abrahão, Rafaela S. Ferreira

**Affiliations:** 1Departamento de Bioquímica e Imunologia, Instituto de Ciências Biológicas, Universidade Federal de Minas Gerais (UFMG), Belo Horizonte 31270-901, Brazil; 2Departamento de Microbiologia, Instituto de Ciências Biológicas, Universidade Federal de Minas Gerais (UFMG), Belo Horizonte 31270-901, Brazilmateusmserafim@gmail.com (M.S.M.S.);

**Keywords:** biochemical assays, NS2B-NS3 protease, scaffold similarity, virtual screening, Zika virus

## Abstract

The NS2B-NS3 protease (NS2B-NS3pro) is regarded as an interesting molecular target for drug design, discovery, and development because of its essential role in the Zika virus (ZIKV) cycle. Although no NS2B-NS3pro inhibitors have reached clinical trials, the employment of drug-like scaffolds can facilitate the screening process for new compounds. In this study, we performed a combination of ligand-based and structure-based in silico methods targeting two known non-peptide small-molecule scaffolds with micromolar inhibitory activity against ZIKV NS2B-NS3pro by a virtual screening (VS) of promising compounds. Based on these two scaffolds, we selected 13 compounds from an initial library of 509 compounds from ZINC15’s similarity search. These compounds exhibited structural modifications that are distinct from previously known compounds yet keep pertinent features for binding. Despite promising outcomes from molecular docking and initial enzymatic assays against NS2B-NS3pro, confirmatory assays with a counter-screening enzyme revealed an artifactual inhibition of the assessed compounds. However, we report two compounds, **9** and **11**, that exhibited antiviral properties at a concentration of 50 μM in cellular-based assays. Overall, this study provides valuable insights into the ongoing research on anti-ZIKV compounds to facilitate and improve the development of new inhibitors.

## 1. Introduction

Zika virus (ZIKV) is an enveloped positive-strand RNA virus transmitted by mosquitoes [1]. ZIKV belongs to the family Flaviviridae and is closely related to other flaviviruses such as dengue virus (DENV), West Nile virus (WNV), yellow fever virus (YFV), and Japanese encephalitis virus (JEV) [2]. In the 2016 outbreak, the World Health Organization (WHO) declared Zika infection a public health emergency due to its association with severe symptoms, neurologic disorders (e.g., Guillain-Barré syndrome), and congenital syndromes, including microcephaly [3,4,5,6]. According to the most recent WHO data from 2022, there are currently 89 countries and territories where ZIKV is still being transmitted at relatively low levels [7]. As of June 2023, no vaccine is available, and no therapeutic option is approved to treat Zika and its infection, with treatment relying upon symptoms’ relief [8]. As a result, efforts have been made to monitor the disease and develop diagnostics, vaccine candidates [8], and therapeutics. Extensive research has been dedicated to understanding and potentially inhibiting ZIKV viral particles’ components, such as structural and nonstructural proteins [9].

Among ZIKV molecular targets, the nonstructural protein NS2B-NS3 protease (NS2B-NS3pro) is a promising target for anti-flavivirus drug design due to its role in proteolysis and viral replication [10,11]. NS2B-NS3pro is the association of the NS3 N-terminal domain, a chymotrypsin-like serine protease with a conserved catalytic triad (His51-Asp75-Ser135), with the hydrophilic region of the membrane-bound NS2B [12]. Available three-dimensional structures of NS2B-NS3pro from ZIKV, DENV, and WNV suggested a functioning mechanism for this target [13,14,15,16,17,18,19,20], while unbound structures have an open and inactive conformation with a flexible and disordered NS2B C-terminal [20]. When bound to a ligand (substrate or inhibitor), the NS2B co-factor wraps around NS3, forming a closed and active conformation, and paired with host cell proteases (e.g., furin), it cleaves the viral polyprotein that is essential for viral assembly and replication [19].

Different studies supported by in silico methods have proposed reversible and irreversible inhibitors with potential broad-spectrum activity targeting ZIKV NS2B-NS3pro [21,22,23]. NS2B-NS3pro inhibitors include repurposed drugs [24,25,26,27,28], substrate-derived peptides [13,17,29,30,31], and small molecules targeting the active site [18,32,33] and allosteric sites [34,35,36]. Although no putative inhibitors have advanced to clinical trials, these drug-like scaffolds can aid in the development of new potential candidates. Thus, drug design efforts can focus on similarity searches based on the concept that comparable molecules can have similar activity [37].

We focused on non-peptide small-molecule inhibitors with micromolar inhibitory activity (IC_50_) and binding affinity (K_d_) against ZIKV NS2B-NS3pro identified by Lee et al. [33]. This set of ten inhibitors (here described as LEE-1 to LEE-10, Figure 1) was categorized into two different scaffolds. Scaffold 1 (LEE-1, -2, -3, -4, -5, -6, -7, -8, and -10) comprises sulfonamide and benzothiazole groups, while scaffold 2 (LEE-9) holds sulfonamide and thiazole connected next to each other [33]. Based on this set of inhibitors, a search in the ZINC15 database for similar commercially available compounds was conducted. Using ligand-based and structure-based in silico methods, we screened the compounds to select those that structurally resemble the two known scaffolds, but with modifications in important groups that differentiate them. A set of 13 compounds was selected and evaluated using biochemical and cellular-based assays. Taken together, our findings offer an understanding of a group of small-molecule inhibitors that may aid in developing potential candidates as ZIKV antivirals.

## 2. Results and Discussion

### 2.1. Using Similarity Search to Build a Compound Library for Virtual Screening

To find ZIKV NS2B-NS3pro inhibitors related to the two scaffolds described by Lee et al. [33], we searched the ZINC15 database [38] (version 2019) for compounds with any structural similarity employing the Substructure and Tanimoto similarity options. The database was queried for both scaffolds, using each of the ten compounds for substructure searches. This search resulted in 509 similar compounds (Appendix A). Using the rCDK package [39], a pair-wise chemical similarity assessment of all compounds was employed against the known inhibitors using the Tanimoto Coefficient (T_c_) [40].

Although most of the known inhibitors belong to scaffold 1 (e.g., conserved sulfonamide, benzothiazole, and phenyl groups), an average T_c_ ≈ 0.51 was determined among them. These highlight different substituents and ring modifications (Figure 1). For instance, the T_c_ ≈ 0.84 among inhibitors LEE-1, LEE-3, and LEE-8 depicts a close similarity and the changes in the terminal group between chlorine, methyl, and methoxy. Therefore, we employed a filter of T_c_ ≤ 0.51 to select compounds that conserved important groups from this class, but that would also contain significant modifications. After applying this filter, we obtained a library of 365 compounds, which presented a chemical space that complements the scaffolds previously identified. The discovery of compounds that possess scaffolds remotely related to those previously reported by Lee et al. [33], while also exhibiting similar activity and drug-like properties, would provide promising results in the search for NS2B-NS3pro inhibitors, as small molecule inhibitors for this target are especially rare. Most of its inhibitors are positively charged and have a high molecular weight [21].

### 2.2. Virtual Screening of Compounds Based on Competitive Inhibitors

To search for novel inhibitors of ZIKV NS2B-NS3pro, we performed molecular docking-based virtual screening (VS) of our library. Since LEE-2 and LEE-3 were reported as competitive inhibitors [33], we focused on the NS2B-NS3pro substrate-binding site. This site contains the catalytic triad, His51, Asp75, and Ser135, and is shallow, solvent-exposed, and negatively charged according to available ZIKV NS2B-NS3pro structures [15]. In a previous study [41], we established a putative binding mode for LEE-2 and LEE-3 in the active site by combining docking and molecular dynamics simulations. These proposed binding modes were stable and anchored by non-covalent interactions, such as aromatic stacking, hydrophobic, and hydrogen bonds with conserved NS2B-NS3pro active site residues [41].

Thus, to preserve this interaction profile, we used DOCK6’s flexible ligand protocol [42] combined with the MultiGrid [43] score function. In this docking approach, a non-bonded interaction energy signature between protein residues and a reference ligand is generated and compared to a candidate signature (i.e., protein residues that interact strongly with the reference ligand) and is used to generate individual grids [43]. These grids allow the algorithm to assess the intermolecular energy profile between the reference and a given ligand pose on each grid, prioritizing poses that interact with the target protein, such as the reference ligand.

Inhibitors LEE-2 and LEE-3 had similar energy signatures (Appendix A) and stable binding modes (Appendix A). Van der Waals and electrostatic interaction energies (kcal/mol) highlighted NS2B residues Ser81* (* indicates NS2B residues), Gly82*, and Asp83*, and NS3 residues His51, Val52, Lys54, Asp75, Pro131, Ser135, Tyr150, Gly151, Asn152, and Tyr161. Some of these residues are conserved and known to be involved in the recognition of substrates and different inhibitors, such as His51, Asp75, Ser135, and Tyr161 in NS2B-NS3pro from other flaviviruses [13,17,18,19]. Also, Asp83* is specific to ZIKV and aids in stabilizing the substrate in its NS2B-NS3pro catalytically active configuration [19,44,45]. Therefore, we performed molecular docking of the 365 compounds guided by both signatures to prioritize binding modes that interact with the same residues.

Poses that predicted binding site complementarity and at least one hydrogen bond with a protein residue were visually inspected [46]. Thus, we selected 36 compounds with scores ranging from −47.70 to −34.95 kcal/mol (Appendix A). Most hydrogen bond interactions occurred between compounds and residues His51, Asp83*, Tyr161, and Ser135. Among this set, we purchased 13 compounds (Table 1, Appendix A), which can be divided into five groups (two clusters and three singletons) based on their scaffolds (Figure 2A) that preserved some elements of scaffold 1 from Lee et al. [33] but have low similarity with them (average T_c_ of 0.34) (Figure 2B).

For instance, all compounds have two phenyl groups linked by a sulfonamide, except compound **8**, in which the sulfonamide nitrogen is replaced with methylene (Table 1). Millies et al. [35] demonstrated the significance of the sulfonamide group in ZIKV inhibitors when an analog lacking this moiety exhibited weak activity. Compounds **8** to **11** have a benzothiazole moiety similar to scaffold 1 from Lee et al. [33]. For compounds **1** to **7** and **12**, a benzimidazole replaces the benzothiazole, distancing them from scaffold 1 from Lee et al. [33]. This inclusion of both benzothiazole and benzimidazole moieties within scaffolds is of interest, as benzothiazole groups are shown as an important structural feature for DENV and ZIKV inhibitors [35,47]. Moreover, benzimidazole-derived compounds were also identified as potential inhibitors of the hepatitis C virus and ZIKV [48,49].

### 2.3. ZIKV NS2B-NS3pro Inhibitory Assays

The selected compounds were subjected to inhibitory assays using a fluorogenic substrate to monitor the ZIKV NS2B-NS3pro proteolytic activity (Table 2). An initial inhibition test was conducted on the compounds at a concentration of 100 µM, except for compounds **8** to **11**, which were tested at 10 µM (the highest soluble concentration in assay conditions). The test was performed with and without a 10 min preincubation of the compounds with the enzyme to detect any time-dependent ligand binding effects (Table 2). Most tested molecules exhibited at least 90% inhibition, regardless of preincubation. Compound **3**, however, exhibited less inhibition (approximately 50% at 100 µM), and among the compounds tested at 10 µM, the highest inhibition was observed for compound **9** (62 ± 8%). It is worth noting that the lack of time dependence observed in the most promising inhibitors suggests that there are no discernible differences in the binding time for this set of related molecules.

We performed an analysis of the half-maximal inhibitory concentration (IC_50_) for the eight compounds that inhibited the enzyme by at least 90% in the initial screening (Figure 3 and Table 2). Notably, compounds **5** and **12** displayed an IC_50_ of 5 µM, while the remaining compounds (**1**, **2**, **4**, **6**, **7**, and **13**) showed IC_50_ values ranging between 13 and 28 µM. It is worth noting that several compounds exhibited high Hill slope values, a phenomenon often observed with aggregators [50]. Such dose–response curves are markedly steeper than those expected for true competitive inhibitors, usually characterized by a Hill coefficient of 1.0 [51]. Except for compound **4**, we detected Hill slope values ranging from 1.5 to 6.7 for all tested compounds (Appendix A), and although high coefficients can stem from other causes, these findings strongly suggest colloidal aggregation.

Confirmatory assays were subsequently assessed to detect eventual false positives among these compounds. The most common cause of artifactual inhibition in enzymatic assays is colloidal aggregation [50,52], which can be formed by some molecules (therefore called aggregators), resulting in promiscuous inhibition. We employed four approaches to detect eventual aggregators among our compounds: changing the detergent present in the assay buffer and its concentration, pre-incubation with bovine serum albumin (BSA), compound evaluation at different enzyme concentrations, and performing an assay against the enzyme cruzain from *Trypanosoma cruzi* (*T. cruzi*), herein referred to as a counter-screening enzyme test.

First, changing the detergent concentration in the assay buffer represents a cost-effective way of obtaining information regarding an aggregator [53]. Increasing detergent concentrations is expected to reduce inhibitory activity by disrupting aggregates, thus suggesting false positives. Here, we investigated the effects of modifying the buffer with 0.001% to 0.01% Triton X-100 on inhibitory activity for all tested compounds (Table 2), and a significant reduction in inhibitory activity was observed when the detergent concentration was increased.

Subsequently, compounds were pre-incubated with BSA before adding the enzyme and initiating the proteolytic reaction [51]. This aims to assess whether BSA could saturate the protein-binding capacity of the aggregates before the addition of ZIKV NS2B-NS3pro, which would cause a reduction in inhibition of the proteolytic activity. Around a 30–70% reduction in inhibition was observed for compounds **1**, **2**, **4**, **6**, and **13** (Table 2), which provide evidence of colloidal aggregation.

Furthermore, we assessed the compounds’ sensitivity to ZIKV NS2B-NS3pro concentration, considering that as enzyme concentration increases, the percentage of inhibition by aggregators tends to decrease [51]. As expected, we observed that all compounds exhibited up to a 60% reduction in inhibitory activity upon a 10-fold increase in enzyme concentration (Table 2).

Last, we also tested the inhibitory activity of the compounds against cruzain. Unfortunately, compounds **6** and **13** were insoluble in the cruzain assay buffer. However, compounds **1**, **2**, and **5** showed 40–100% inhibition against the cysteine protease, as expected for promiscuous inhibitors such as aggregators [51]. Conversely, no inhibitory activity of compound **4** was observed against cruzain, suggesting a specific inhibition of ZIKV NS2B-NS3pro. Despite this specificity, aggregation is a condition-dependent phenomenon [50], and while the detection of promiscuous inhibition is a strong indication of aggregation, it is also possible to observe specificity because of the lack of aggregation in an assay buffer.

Altogether, these results show that the inhibition observed is due to colloidal aggregation, which reinforces the importance of careful compound validation in biochemical assays, even when working with a series related to literature leads. In a parallel study, we also evaluated the compounds LEE-2 and LEE-3 after resynthesis and verified that any inhibition observed was due to colloidal aggregation [54]. Similarly, for a series of thiosemicarbazones with potent trypanocidal activity recently reported [55], it was demonstrated that cruzain inhibition at low nanomolar concentrations was due to colloidal aggregation. It is worth noting that thiosemicarbazones are well-established as potent cysteine protease inhibitors [56]. Aggregators were also identified among a series of competitive cruzain inhibitors, either solely responsible for enzyme inhibition or resulting in a dual mechanism of inhibition [55,57,58], which also highlights the need for continued monitoring for possible false positives while attempting to optimize bioactive compounds.

Finally, we highlight the importance of experimentally investigating aggregation. It is tempting to develop computational models to predict aggregation, and models with reasonable accuracy [59,60] have been developed. However, it is also known that even drugs successfully used in the clinic can aggregate at high concentrations [61]. Considering the concentration- and buffer-dependency [50,61] reported for colloidal aggregators, it is essential to experimentally investigate if the compound of interest aggregates at concentrations similar to those in which biochemical inhibition is observed.

### 2.4. Antiviral Activity Evaluation against ZIKV

In parallel with biochemical validation, the selected set of 13 compounds was assessed with a single MTT assay in 96-well microplates. Compounds showed CC_50_ values ranging from <12.5 to 119.79 ± 3.68 μM (Table 3). The effective concentration of 50% (EC_50_) was also assessed with MTT (same conditions as previous), adding a viral suspension (MOI of 0.1) of ZIKV (PE243). Here, two compounds, **9** and **11**, showed antiviral activity at 50 μM, resulting in selectivity indexes (SI) of 2.02 and 1.47, respectively. These compounds that showed antiviral activity could be evaluated in the enzyme assays only at concentrations up to 10 μM, due to their low solubility in the assay buffer. It is also important to consider that experimental determination of inhibitors should follow in vitro determination in cellular-based assays (e.g., cytotoxicity and antiviral activity) [62,63], and those may complement initial computational approaches and target validation, such as the proposed ZIKV NS2B-NS3pro inhibitors presented in this study. Therefore, further experiments are needed to determine the mode of action of compounds **9** and **11**.

## 3. Materials and Methods

### 3.1. Similarity Search Approach

We queried the ZINC15 database (version 2019) [38] to retrieve compounds with similar structures and substructures. Specifically, the SMILES representation of the ten known inhibitors (IC_50_ < 50 µM), from Lee et al. [33], were individually used as queries employing the Substructure and Tanimoto similarity search options to find commercially available compounds. The scaffold of these inhibitors incorporates benzothiazole, thiazole, and sulfonamide moieties, relevant to ZIKV NS2B-NS3pro inhibitors [35]. ZINC15 results were saved in SMILES format. SMILES were converted into ECFP [65] fingerprints with the R package rCDK (version 2.4.0) [39]. The Tanimoto coefficient (T_c_) was used to compare compounds’ fingerprints, forming a matrix with values from 0 (no similarity) to 1 (identical) [66]. Thus, higher T_c_ values show more similarity between compounds, and a molecule compared to itself has a T_c_ of 1. We selected compounds that had T_c_ ≤ 0.5 with all the ten known inhibitors for molecular docking. Thus, we expected compounds that were different but maintained the scaffolds of the known inhibitors.

### 3.2. Dataset Preparation

When available, three-dimensional structures were obtained directly from ZINC15 [38] in the MOL2 format. The ones with only SMILES format were converted to three-dimensional structures with OpenBabel (version 2.3.2) [67] and Avogadro (version 1.2.0) [68]. All compounds were minimized with 100 steps of the steepest descent algorithm to attain low-energy conformations using OpenBabel (version 2.3.2) [67]. AM1-BCC partial charges were assigned using the Antechamber module [69] for all compounds.

### 3.3. Virtual Screening by Molecular Docking

Molecular docking was performed with DOCK6 (version 6.8) [70] with the flexible ligand protocol described in Mukherjee et al. [42] paired with the MultiGrid score [43]. Previously in Santos et al. [41], we refined and relaxed structures of ZIKV NS2B-NS3pro complexed to LEE-2 and LEE-3 [33] using molecular docking and molecular dynamics. Thus, in this work, the atomic coordinates of these complexes were employed as a reference, preserving the hydrogen atoms and protonation states of the standard receptor residues from Santos et al. [41]. Catalytic residues His51 and Ser135 were both neutral, with the His51 tautomer protonated at Nδ (HID), while Asp75 was deprotonated, and all remaining aspartic and glutamic acids were also preserved as deprotonated. Protein atomic partial charges from the AMBER FF14SB [71] force field were kept. The final compounds were selected based on docking scores, protein–ligand interactions, and overall binding site complementarity by visual inspection [46].

### 3.4. NS2B-NS3pro Expression and Purification

The expression and purification protocol used to obtain the ZIKV NS2B-NS3pro was first described by Lei et al. [19], and the same group kindly provided the construct in a pET-15b plasmid. The construct codes for NS2B-NS3pro come from the Brazilian isolate BeH823339, with a Gly4-Ser-Gly4 linker between the two protein chains and four-point mutations: R96A (in NS2B) and R29G/C80S/C143S (in NS3) (SisGen accession code: ACCD10D).

BL21-DE3 cells were transformed using this plasmid and were grown overnight (12 h) at 37 °C and 200 rpm of constant agitation in 12.5 mL of 2xYT sterile media containing 100 μg/mL of ampicillin. Next, the grown media was added to 1 L of media under the same conditions. After achieving an optical density of 0.7 ± 0.1, protein overexpression was inducted with 1 mM of isopropyl-β-D-galactoside (IPTG), and the culture was kept overnight at 20 °C and 200 rpm of constant agitation. Next, the inducted media was centrifuged for 30 min at 5000 rpm at 4 °C, and the cells were resuspended using 20 mL of buffer A (25 mM Tris-HCl, 10 mM NaCl, 5% glycerol, pH 8.5). Finally, the cells were lysed via sonication under on/off pulses of 20/40 s in ice and centrifuged at 4 °C and 10,000× *g* for 1 h.

The supernatant was injected into a 5 mL HisTrap Sepharose HP (GE Healthcare, Chicago, IL, USA) nickel column. First, the ZIKV protease was eluted at a flow rate of 1 mL/min of buffer and a linear gradient covering 5 column volumes from 0 to 100% of buffer B (25 mM Tris-HCl, 500 mM NaCl, 500 mM imidazole, 5% glycerol, pH 8.5). Next, the protein solution was loaded into a HiLoad 16/600 Superdex 75pg (GE Healthcare) gel-filtration column and eluted using buffer A at a flow rate of 0.1 mL/min by 1.2 column volumes. The resulting protein samples were stored at −80 °C.

### 3.5. NS2B-NS3pro Enzyme Assays

To evaluate the ZIKV protease complex’s activity cleavage of a modified peptide (Bz-Nle-Lys-Lys-Arg-AMC) was monitored using the fluorescent group 7-Amino-4-Methylcoumarin (AMC) combined with a benzoyl (Bz) quencher group. Fluorescence from the substrate cleavage was measured for 10 min using a microplate spectrofluorometer (Biotek Synergy 2, Winooski, VT, USA), with an excitation wavelength of 340 nm and an emission wavelength of 440 nm. These experiments were conducted on 96-well flat-bottom black microplates. A linear adjustment technique was employed to establish the initial reaction velocity of the potential inhibitors, which were compared to a control containing DMSO. Each condition was tested in triplicate and two independent assays (n = 6 data points).

Before the assays against the enzyme, the compounds’ solubility when dissolved in the assay buffer at a concentration of 100 μM was evaluated by visual inspection of transparent microplates and microtubes. Compounds **8**–**11** were insoluble at 100 μM and were therefore tested at their highest soluble concentration (10 μM). During the preliminary screening phase, the compounds underwent two distinct assays: one with a 10 min pre-incubation period with the enzyme and one with no pre-incubation step. Both assays occurred in a buffer solution comprising 10 mM Tris-HCl at pH 8.5, 0.005% Tween-20, and 5% glycerol. The final assay conditions consisted of an enzyme concentration of 0.2 nM and a substrate concentration of 44 µM. Compounds that exhibited a 90% or higher inhibition of the enzyme’s activity at 100 μM underwent a concentration-response inhibition assay under the same buffer conditions to determine their IC_50_ values. Nonlinear regression analysis employing a four-parameter logistic curve with a variable slope was used to calculate concentration-response curves. GraphPad Prism 6.0 software (GraphPad Software, San Diego, CA, USA) was used for data analysis.

Confirmatory assays of promising compounds were conducted by adapting well-established assays [51]. Briefly, to assess the compounds’ sensitivity to detergent, we determined percentages of enzyme inhibition in the presence of 0.001% and 0.01% Triton X-100, without the addition of Tween-20, after a 10 min pre-incubation of the compounds with the enzyme. In a second assay, we evaluated the effect of compound pre-incubation with bovine serum albumin (BSA) for 10 min, followed by the addition of the protease and another 10 min incubation before the addition of the substrate solution to start the enzymatic reaction. For this assay, the final concentrations of substrate and enzyme and buffer conditions were maintained the same as those used in the initial screening, while the final BSA concentration was set to 1 mg/mL. Last, we evaluated the impact of enzyme concentration on the assays with the protein detergent CHAPS 1 mM and each compound at a concentration close to its IC_50_. The assay was performed by varying the NS2B-NS3pro concentration (0.2 nM and 2 nM). The compounds were pre-incubated with the enzyme for 10 min at 37 °C.

### 3.6. Cruzain Enzyme Assays

Cruzain activity was measured by monitoring the fluorescence signal obtained by cleavage of the fluorogenic substrate Z-Phe-Arg-AMC, following a well-established protocol [55,57,72,73]. The fluorescence intensity (excitation λ: 340 nm; emission λ: 440 nm) was monitored for at least 5 min. All assays were performed in sodium acetate 0.1 M at pH 5.5 and β-mercaptoethanol 10 mM. The final protein concentration was 0.1 nM, and the substrate concentration was 2.5 µM. Each condition was tested in triplicate and two independent assays (n = 6 data points), with preincubation of the enzyme for 10 min at 25 °C. Assays were performed at compound concentrations near the IC_50_ value against ZIKV NS2B-NS3pro.

### 3.7. Cell Lineage and Virus Strain

Viruses were kindly provided by the Laboratório de Vírus at UFMG, Brazil. Zika virus (ZIKV) PE243 was obtained as described by Donald et al. [74]. The Vero (ATCC^®^ CCL-81™) cell line was used for the cytotoxicity assays and antiviral activity evaluation against ZIKV. Cells were cultured in Eagle’s minimum essential media (MEM) (Cultilab, Campinas, Brazil). The media was supplemented with 5% fetal bovine serum (FBS) (Cultilab, Brazil), in addition to 100 IU/mL penicillin (Cellofarm, Rio De Janeiro, Brazil), 100 μg/mL streptomycin (Merck, Darmstadt, Germany), and 0.25 μg/mL amphotericin B (Cultilab, Brazil). This work is registered in SisGen under number AA7854F.

### 3.8. Viral Propagation

Viral propagation was conducted according to Serafim et al. [64]. Vero monolayers with 60–80% confluence were washed twice with phosphate-buffered saline (PBS) and infected at a multiplicity of infection (MOI) of 0.01 for ZIKV, approximately one viral particle per 100 cells. Adsorption was made for 1 h at 37 °C in a 5% CO_2_ atmosphere with 2 mL of diluted virus in MEM (without FBS), gently homogenizing the flasks every 10 min. After adsorption, 12 mL of MEM with 2% FBS was added to the flasks and incubated under the same conditions. Cell monolayers were observed daily under optical microscopy until the cytopathic effect was approximately 80%. The supernatant was removed and centrifuged at 2016× *g* in an RT6000B centrifuge (Sorvall, Thermo Scientific, Waltham, MA, USA) for 10 min at 4 °C. Viruses were stored at −70 °C.

### 3.9. Viral Titration

Following the previous study by Serafim et al. [64], viral titration experiments were performed. Cells seeded in 24-well microplates (8.0 × 10^5^ cells per well) were incubated at 37 °C in a 5% CO_2_ atmosphere for 24 h. Each well received 100 μL of ZIKV’s serial dilutions in MEM in a 1:10 ratio (10^−1^ to 10^−5^), following adsorption for 1 h. The cells were overlayed with 1.0 mL of 199 media (Cultilab, Brazil) with 2% FBS and 1% carboxymethylcellulose (CMC) (Synth, Diadema, Brazil). Microplates were incubated for five days at 37 °C in a 5% CO_2_ atmosphere and then fixed with 10% formalin overnight, gently washed with distilled water, and stained with a 1% crystal violet solution for 20 min [75].

### 3.10. Cytotoxicity Assay: 50% Cytotoxic Concentration (CC_50_)

Cells were seeded in 96-well microplates (4.0 × 10^4^ cells per well) and incubated at 37 °C and 5% CO_2_ for 24 h. Then, 200 µL of MEM with 1% FBS containing a serial dilution of the compounds (100 to 12.5 µM) was added. As vehicle control, a serial dilution of DMSO was used. After 72 h of incubation under the same conditions, 100 μL of 3-(4,5-dimethylthiazol-2-yl)-2,5-diphenyltetrazolium bromide (MTT; ThermoFischer Scientific, Waltham, MA, USA) [76] diluted in MEM (0.5 mg/mL) was added to each well and incubated for 3 h. The media was removed, and 100 μL of DMSO was added to each well to solubilize formazan crystals and shaken for 20 min. The absorbance was read at 570 nm using a spectrophotometer (VersaMax, Molecular Devices, San Jose, CA, USA). The percentages of inhibition of cell viability were calculated as the ratio between the absorbance of the compound-treated cells and cells with only the vehicle. The cytotoxic concentration of 50% (CC_50_) is defined as the lowest concentration of a specific compound that reduces by 50% the viability of cultured cells. Linear regression (LR) was used for the analysis, considering results with r^2^ > 0.9. All conditions were tested in triplicate and one independent assay (n = 3 data points).

### 3.11. Antiviral Activity Assay: 50% Effective Concentration (EC_50_)

Cells were seeded in 96-well microplates (4.0 × 10^4^ cells per well) and incubated at 37 °C and 5% CO_2_ for 24 h. Then, 100 µL of MEM with 1% FBS containing a serial dilution of the compounds below their CC_50_ values (ranging from 100 to 6.25 µM) was added together with 100 μL of the viral suspensions (MOI of 0.1) in MEM with 1% FBS. Wells that contained only viruses were used as infection control, with a serial dilution of DMSO. Treatment with MTT follows the same protocol as for CC_50_. The EC_50_ was calculated as the percentage of the ratio between the absorbance of the compound-treated infected cells and infected cells with vehicle only. Ribavirin was added as a comparison control. LR was also used for the analysis, considering results with r^2^ > 0.9. All conditions were tested in triplicate in one independent assay (n = 3 data points).

## 4. Conclusions

We used a ligand-based in silico approach to screen compounds that structurally resembled ten known competitive inhibitors of NS2B-NS3pro but contained significant modifications. The compounds were then submitted to a structure-based molecular docking protocol that guided and prioritized binding modes interacting with important residues such as His51, Asp75, Ser135, Tyr161, and Asp83*. We selected 36 compounds exhibiting binding site complementarity and specific interactions with these residues based on visual inspection and their score-based affinity. Thirteen compounds were then purchased and submitted to experimental validation, in which enzyme inhibition was observed because of aggregator characteristics. These findings highlight the crucial importance of confirmatory tests in detecting artifacts in the experimental screenings of compounds. False positives in enzymatic assays are frequently caused by aggregation, yet this issue is often overlooked in studies in this field. Nonetheless, it is worth noting that two compounds, termed compounds **9** and **11**, exhibited an EC_50_ of 50 μM in antiviral assays with SI values of 2.02 and 1.47, respectively. Additional research is needed to study their mode of action since they showed limited solubility in the assay buffer and could not be extensively characterized in enzymatic assays. Taken together, these findings contribute to a better understanding of the behavior of serine proteases and their inhibitors and may aid in the design, discovery, and especially development of potential lead candidates for the treatment of Zika.

## Figures and Tables

**Figure 1 pharmaceuticals-16-01319-f001:**
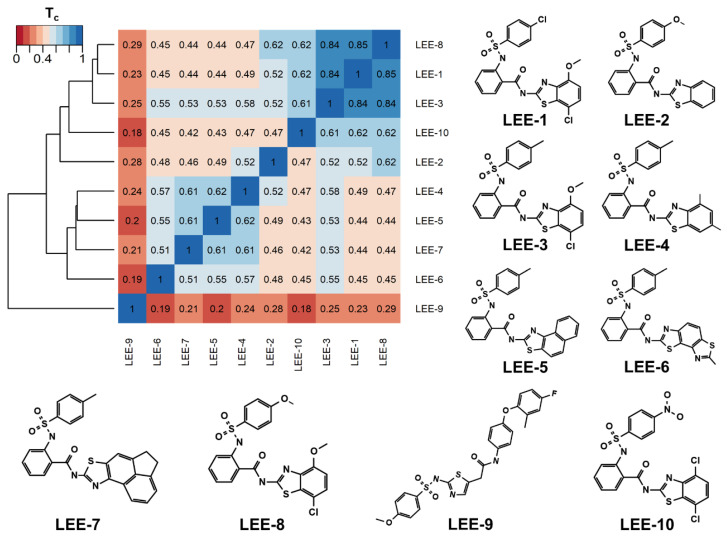
Heatmap of LEE-1-10’s Tanimoto Coefficients (T_c_). The ten known inhibitors from Lee et al. [33] are colored from low (red) to high (blue) similarity. The hierarchical clustering dendrogram (left) shows the scaffold division between scaffold 1 (LEE-1, -2, -3, -4, -5, -6, -7, -8, and -10) and scaffold 2 (LEE-9).

**Figure 2 pharmaceuticals-16-01319-f002:**
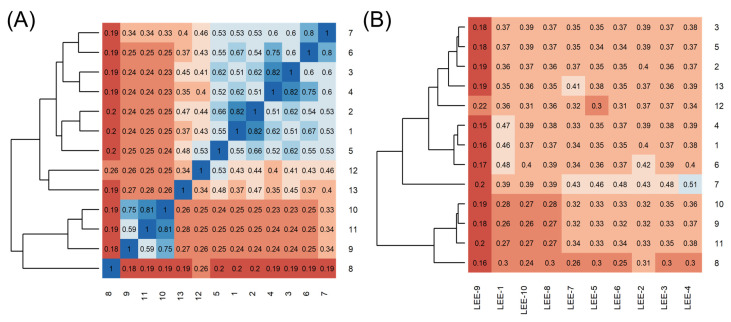
Heatmaps of T_c_ for the chosen 13 compounds. Heatmaps are colored from low (red) to high (blue) similarity. (**A**) T_c_ comparison heatmap of the 13 selected compounds. According to the hierarchical clustering dendrogram (left), compounds **1** to **7**, **8** (singleton), **9** to **11**, **12** (singleton), and **13** (singleton) form five clusters based on their scaffolds. (**B**) T_c_ comparison between the ten known inhibitors from Lee et al. [33] and the chosen 13 compounds.

**Figure 3 pharmaceuticals-16-01319-f003:**
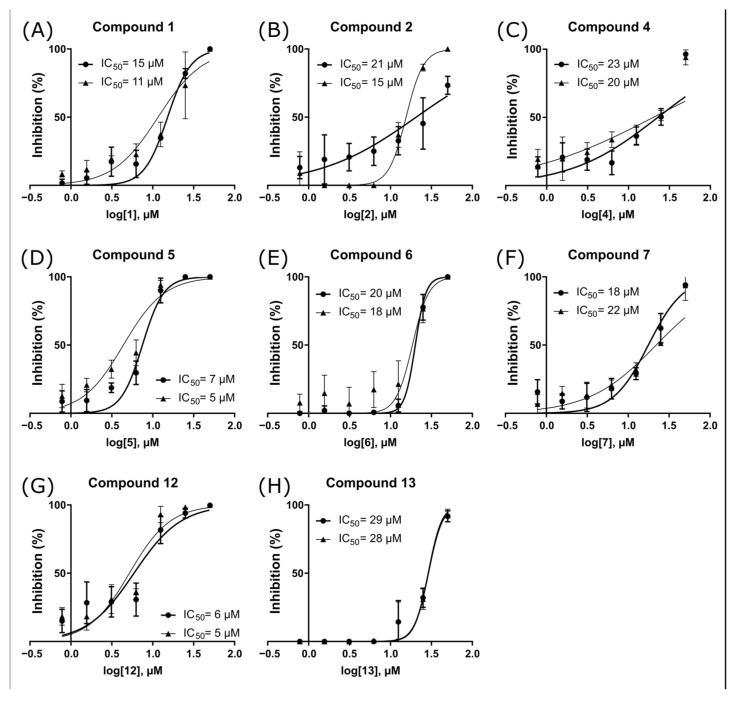
IC_50_ curves for NS2B-NS3pro inhibitors. Curves for compounds 1 (**A**), 2 (**B**), 4 (**C**), 5 (**D**), 6 (**E**), 7 (**F**), 12 (**G**), and 13 (**H**) are represented. Each curve was based on the percentage of ZIKV NS2B-NS3pro inhibition in the presence of seven compound concentrations. Each condition was assessed in triplicate and repeated in two independent assays (*n* = 6 data points).

**Table 1 pharmaceuticals-16-01319-t001:** Final compound selection to be tested against ZIKV NS2B-NS3pro.

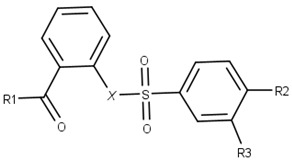
**Compound**	**R1**	**R2**	**R3**	**X**
**1**	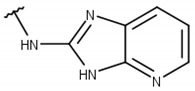	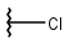	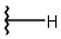	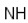
**2**	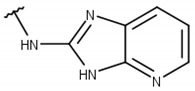	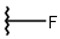	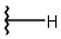	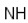
**3**	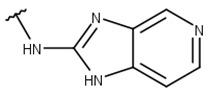	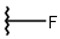	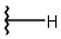	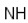
**4**	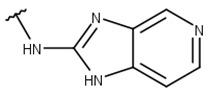	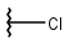	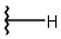	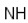
**5**	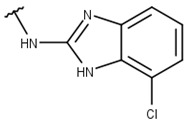	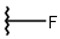	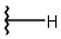	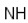
**6**	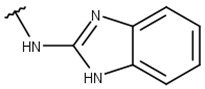	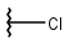	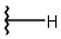	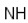
**7**	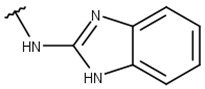	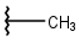	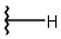	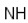
**8**	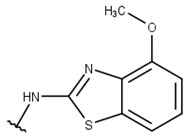	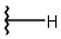	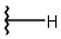	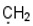
**9**	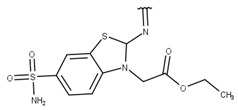	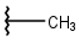	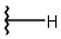	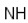
**10**	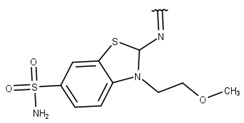	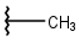	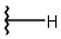	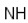
**11**	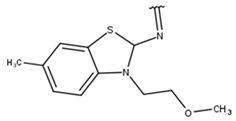	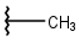	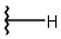	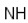
**12**	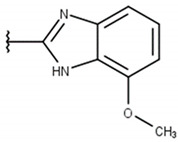	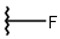	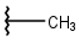	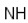
**13**	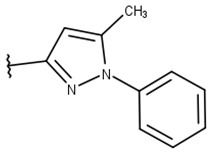	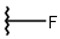	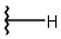	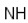

**Table 2 pharmaceuticals-16-01319-t002:** Final compound selection to be tested against ZIKV NS2B-NS3pro.

	% ZIKV NS2B/NS3pro Inhibition (100 µM)					ZIKV NS2B/NS3pro IC_50_ (µM)	% ZIKV NS2B/NS3pro Inhibition		% Cruzain Inhibition ^b^
Compound	With Preincubation	Without Preincubation	0.001% Triton	0.01% Triton	BSA		[Enzyme] = 0.2 nM ^b^	[Enzyme] = 2 nM ^b^	
**1**	100 ± 0	100 ± 0	93 ± 8	47 ± 34	69 ± 6	13 ± 3	45 ± 0.1	26 ± 7	97 ± 7
**2**	100 ± 0	100 ± 0	94 ± 7	30 ± 19	32 ± 2	18 ± 4	30 ± 0.1	28 ± 7	42 ± 11
**3**	57 ± 22	56 ± 11	ND	ND	ND	ND	ND	ND	ND
**4**	100 ± 0	100 ± 0	93 ± 8	25 ± 15	45 ± 4	21 ± 2	44 ± 0.2	29 ± 8	−12 ± 9.1 ^c^
**5**	100 ± 0	100 ± 0	93 ± 3	6 ± 8	94 ± 8	5 ± 3	62 ± 0.1	23 ± 8	69 ± 24
**6**	100 ± 0	99 ± 2	89 ± 6	7 ± 9	55 ± 20	19 ± 2	99 ± 0.4	66 ± 21	ND
**7**	100 ± 0	90 ± 8	88 ± 5	12 ± 17	89 ± 7	19 ± 4	87 ± 0.04	27 ± 6	ND
**8** ^a^	8 ± 8	15 ± 11	ND	ND	ND	ND	ND	ND	ND
**9** ^a^	62 ± 8	58 ± 8	90 ± 3	12 ± 7	0	ND	ND	ND	ND
**10** ^a^	42 ± 3	30 ± 12	87 ± 3	8 ± 3	0	ND	ND	ND	ND
**11** ^a^	13 ± 4	9 ± 7	ND	ND	ND	ND	ND	ND	ND
**12**	100 ± 0	91 ± 9	90 ± 5	5 ± 7	87 ± 18	5 ± 1	94 ± 0.06	30 ± 10	ND
**13**	95 ± 2	100 ± 1	100 ± 0	45 ± 9	57 ± 15	28 ± 1	43 ± 0.1	26 ± 10	ND

^a^ Compounds evaluated at 10 due to solubility limitations ^b^ Compounds were evaluated at a concentration close to their IC_50_ values against ZIKV NS2B/NS3pro: 25 µM in the case of compounds **1**, **2**, **4**, **6**, **7**, and **13**; 10 µM for compounds **5** and **12**. ^c^ Enzyme activation was observed instead of enzyme inhibition. ND = not determined

**Table 3 pharmaceuticals-16-01319-t003:** Biological evaluation of the selected compounds against ZIKV (PE243).

Compound	CC_50_ (µM)	EC_50_ (µM)
**1**	22 ± 0.79	NA
**2**	32.47 ± 1.95	NA
**3**	119.79 ± 3.68	NA
**4**	57.11 ± 1.86	NA
**5**	<12.5	NA
**6**	38.36 ± 2.18	NA
**7**	32.76 ± 1.55	NA
**8**	23.57 ± 1	NA
**9**	100.95 ± 4.23	50
**10**	76.21 ± 3.81	NA
**11**	73.36 ± 4.02	50
**12**	<12.5	NA
**13**	<12.5	NA
Ribavirin ^a^	>100	4.1 ± 0.35

^a^ Results from Serafim et al. [64]. NA = not active.

## Data Availability

Data are contained within the article and the Appendix A.

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
