# Peer review of "Evaluating Known Zika Virus NS2B-NS3 Protease Inhibitor Scaffolds via In Silico Screening and Biochemical Assays"

_pharmaceuticals, 2023, doi:10.3390/ph16091319_

Round 1
Reviewer 1 Report
Santos et al. evaluated Zika virus NS2B-NS3 protease inhibitor scaffolds by in silico screening and biochemical assays. The topic is very interesting, but I have a couple of issues that need to be addressed.
- There is a big problem in the references: sometimes the authors used a number, and sometimes they used a family name and year. For example, lines 79, 137, 149, 152, 154, 291, 296, and 306. References should be unique according to the instructions of the journal.
- There are many grammar mistakes in the whole manuscript, and it should be carefully revised.
- In order to fulfill the biochemical assay, authors should provide antimicrobial data for the selected compounds.
- To find ZIKV NS2B-NS3pro inhibitors related to the two scaffolds described by Lee et al. (2017), we searched the ZINC15 [38] database for compounds with any structural similarity, authors must mention the searching method.
Author Response
Dear reviewer, thank you very much for all comments and suggestions. Please find attached a point-by-point response.
Sincerely,
Profa. Rafaela Ferreira

Reviewer 2 Report
The manuscript describes a screening by the authors, using both computational methods and experimental methods, on compounds that have the potential to be the inhibitors for NS2B-NS3 protease, a therapeutic target for Zika virus-related diseases. This work demonstrates the need for counter screening to exclude false positive results in a screening process.
I have a few minor suggestions:
1. it appears that the need for counter screening originates from the biochemical assays that are used in the screening process. It would be of much higher significance if the authors can provide a discussion on how to best eliminate the false positives in the computational screening round (prior to the experimental screening).
2. The work from Lee et al. seems to have a huge impact on the design of this work. Can the authors elaborate more on the structural similarity and the related antiviral mechanisms of their own compound and the compound reported by Lee et al.?
Format check on superscript/subscript is needed, e.g., "CO2" in line 410, "4.0 x 104" cells per well in line 415 and 432.
Author Response

(The authors gave the same response as above.)

Round 2
Reviewer 1 Report
The authors addressed all of my inquiries. I recommend that the paper be accepted.